# ViDROP: Video Dense Representation through Spatio-Temporal Sparsity

## Abstract

Self-supervised learning (SSL) has revolutionized image processing, but extending its success to video understanding presents unique challenges due to increased data complexity and computational demands. We introduce ViDROP (Video Dense Representation thrOugh spatio-temporal sParsity), a novel SSL architecture for video understanding that combines token dropping and masking strategies. Our approach eliminates the need for a decoder and enables per-patch loss computation, overcoming limitations of previous video SSL methods. Moreover, we propose a simple yet effective video compression technique using k-means clustering in pixel space, significantly accelerating data loading and facilitating rapid experimentation. ViDROP demonstrates remarkable scalability across model sizes, from ViT-Small to ViT-Huge, when starting from pretrained models (VideoMAE or V-JEPA), achieving significant performance gains. Pushing the boundaries even further, we leverage network expansion techniques to successfully train ViT-Huge from scratch using modest computational resources, achieving comparable accuracy to VideoMAE 25× faster in training time. This marks a significant breakthrough in large-scale video SSL, enabling the training of state-of-the-art models with limited resources. Extensive experiments show that ViDROP achieves state-of-the-art performance on various video understanding benchmarks, including Kinetics400, SSv2, UCF101, and HMDB51, as well as in temporal action detection (THUMOS14). These results highlight the effectiveness of our fine-grained token-level learning strategy in a domain traditionally dominated by fine-tuned SSL models, while enabling the training of large-scale models with limited computational resources.

## 1 Introduction

Self-supervised learning (SSL) has transformed computer vision by enabling models to learn rich representations from vast amounts of unlabeled data. While SSL methods like DINOv2 Oquab et al. (2023) have achieved remarkable success in image processing, extending these techniques to video understanding presents unique challenges due to increased data complexity and computational demands.

The temporal dimension in videos captures essential information about object movement and scene changes, crucial for action understanding. However, it also significantly increases the amount of data to be processed, exacerbating computational burdens. Recent works such as VideoMAE Tong et al. (2022) have addressed these challenges by employing sparse encoder and dense decoder architectures, leveraging vision transformers Dosovitskiy et al. (2020) to efficiently process video data split into tokens or patches.

Despite their promise, these approaches face two significant limitations. First, the computational cost of the decoder remains considerable, as it still operates on all or a large portion of video tokens (see *e.g.*, Wang et al. (2023a); Hwang et al. (2022)). Second, the split between encoder and decoder, coupled with a low-level reconstruction objective, can lead to suboptimal representation learning. The encoder may not capture all relevant high-level features, as some information is relegated to the decoder for reconstruction purposes. This division of representational power can result in less robust or comprehensive features, potentially misaligning with high-level downstream tasks.

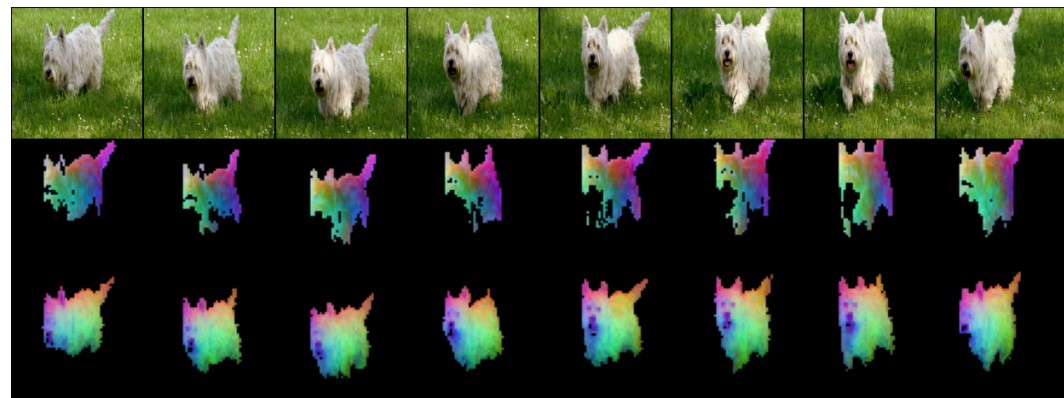

Figure 1: **Visualization of per-patch representations.** Top row: Original video frames of a dog walking on grass. Middle row: 3-channel PCA visualization Oquab et al. (2023) of patch features from VideoMAE Tong et al. (2022). Bottom row: 3-channel PCA visualization of patch features from ViDROP. The more detailed and coherent colorization in the bottom row demonstrates ViDROP's ability to produce higher-quality per-patch representations, capturing finer details and maintaining better consistency (semantic correspondences) across frames.

To address these challenges, we propose ViDROP (Video Dense Representation thrOugh spatio-temporal sParsity), a novel SSL architecture for video understanding that combines token dropping and masking strategies. ViDROP employs a sparse encoder with masked tokens and eliminates the need for a decoder, enabling per-patch loss computation for robust representations while maintaining exceptional efficiency. This approach overcomes the limitations of previous methods, providing a more effective and computationally efficient solution for self-supervised learning in video understanding tasks (see Figure 2).

To accelerate training, we introduce a simple yet effective video compression technique using k-means clustering Lloyd (1982) in pixel space. This innovation addresses the data loading bottleneck that emerges as our model accelerates computation, especially for smaller models.

ViDROP achieves state-of-the-art linear probing results using only minimal data augmentations (random resized cropping and flipping), in contrast to top-performing models like SVT Ranasinghe et al. (2021) and $\rho$BYOL Feichtenhofer et al. (2021) that rely on heavy, potentially dataset-specific augmentations. This approach aims for more robust and generalizable representations. Additionally, ViDROP produces high-quality per-patch representations alongside strong global representations, distinguishing it from traditional contrastive methods that excel in global video clip representations but may struggle with fine-grained, per-patch understanding (see Figure 1). This versatility enables ViDROP to be effective across a wide range of downstream tasks, from those requiring local understanding to those needing global video comprehension.

Our contributions can be summarized as follows:

- **A novel, efficient SSL architecture for video understanding:** We introduce ViDROP, combining sparse token processing with masked learning, achieving state-of-the-art performance without a decoder while producing high-quality per-patch and global representations.

- **Scalable and resource-efficient training:** We demonstrate exceptional scalability across model sizes (ViT-Small to ViT-Huge) and initialization strategies (VideoMAE Tong et al. (2022), V-JEPA Bardes et al. (2024)). By leveraging network expansion techniques Wang et al. (2023b) and introducing a novel k-means clustering-based video compression method, we train a ViT-Huge model with comparable accuracy to VideoMAE in just 4% of the total training time, **enabling large-scale video SSL with limited computational resources.**

- **Comprehensive evaluation:** ViDROP achieves superior performance across a wide range of video understanding tasks, including action recognition, temporal action detection, frame-wise tracking, and copy detection. Our method's versatility extends to image classification, showcasing the robustness and generalizability of the learned representations.

## 2 PRIOR WORK

**Evolution of image SSL:** Self-supervised learning (SSL) in computer vision has evolved from image-based to video understanding approaches. Image SSL progressed from contrastive learning van den Oord et al. (2018); Chen et al. (2020a;b) to clustering-based methods Caron et al. (2018; 2020; 2021), then to direct feature prediction Grill et al. (2020); Chen & He (2020) and cross-correlation techniques Zbontar et al. (2021). These discriminative approaches have consistently advanced representation learning in static images.

**Masked modeling and diverse SSL approaches:** Complementing these discriminative methods, various other approaches have played a crucial role in SSL. Notably, masked modeling approaches have gained significant traction. BEiT Bao et al. (2021) introduced the concept of infilling in latent space, inspired by BERT Devlin et al. (2019) in natural language processing. Masked Autoencoders (MAE) He et al. (2021) adapted this idea to regress missing pixel patches directly in pixel space, while AIM El-Nouby et al. (2024) advanced these concepts with next token prediction in latent space. Hybrid models such as iBOT Zhou et al. (2021) and DINOv2 Oquab et al. (2023) have further pushed the boundaries, combining generative and discriminative elements to produce high-quality representations. These approaches have demonstrated exceptional performance in tasks like KNN classification Fix & Hodges (1989) and unsupervised semantic segmentation Hamilton et al. (2022), showcasing the power of integrating multiple SSL paradigms.

**Video SSL challenges:** The transition from image to video SSL introduces unique challenges due to the temporal dimension and increased data scale. Video SSL methods often employ pretext tasks designed to capture both spatial and temporal aspects, such as determining the correct order of shuffled video frames Misra et al. (2016); Jenni et al. (2020); Dorkenwald et al. (2022); Dave et al. (2023). Generative models like VideoMAE Tong et al. (2022) and VideoMAEv2 Wang et al. (2023a) have been adapted to reconstruct video segments or entire frames, while consistency-based approaches like $\rho$BYOL Feichtenhofer et al. (2021), BRAVE Recasens et al. (2021), and SVT Ranasinghe et al. (2021) ensure feature consistency across different video segments.

**Computational optimization:** The computational demands of processing videos, especially with Vision Transformers (ViT) Dosovitskiy et al. (2020), have led to various optimization strategies. Factorized attention, as used in SVT Ranasinghe et al. (2021), offers one approach, although comprehensive spatio-temporal attention, as in ViViT Arnab et al. (2021), typically yields superior results. Some methods leverage pretrained image or short video clip encoders to derive representations for longer clips Sameni et al. (2022); Lei et al. (2021). Inspired by MAE He et al. (2021), sparse encoder inputs paired with dense decoder outputs have been employed to reduce computational requirements, with models like VideoMAEv2 Wang et al. (2023a) and EVEREST Hwang et al. (2022) further economizing resources by introducing sparse reconstruction tokens to the decoder.

**Data processing bottlenecks:** Addressing the data bottleneck in video SSL remains a significant challenge. While tools like FFCV Leclerc et al. (2023) and its SSL variant Bordes et al. (2023) are effective for images, they fall short for video processing. Wrappers such as decord and Avion Zhao & Krahenbuhl (2023) offer some improvements but remain relatively slow. Data remasking techniques Bardes et al. (2024); Feichtenhofer et al. (2022) provide a cost-effective way to alleviate data loading bottlenecks. Another approach involves using precomputed latent representations of videos Wiles et al. (2022); Jaegle et al. (2021), which can be integrated with learned data augmentations Lee et al. (2023). However, this method introduces higher costs at the inference stage due to the need for an expensive encoder to convert raw pixel videos into the latent representation.

**Role of data augmentation:** The role of data augmentation in SSL has been a subject of significant research Zhai et al. (2023); Wagner et al. (2022); Kalibhat et al. (2023). While heavy augmentations have been crucial for the success of many image SSL methods Chen et al. (2020a); Grill et al. (2020), their application to video data is more complex due to the need to preserve temporal coherence Feichtenhofer et al. (2021). Some video SSL approaches have adapted image augmentation techniques to the video domain Ranasinghe et al. (2021), while others have developed video-specific augmentations Qian et al. (2020). However, the computational cost of these augmentations can be substantial, especially for large-scale video datasets. Moreover, the reliance on carefully designed augmentations raises questions about the broader applicability and robustness of SSL methods across diverse video datasets and tasks.

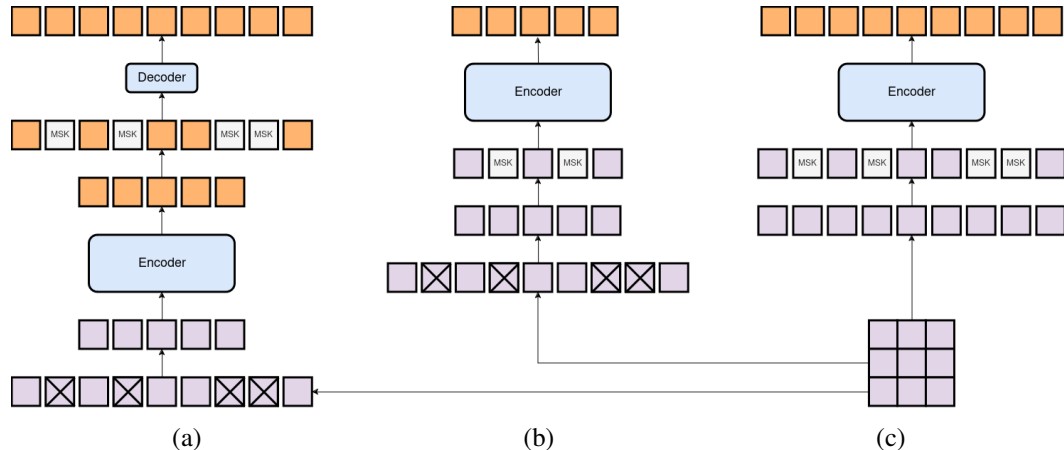

Figure 2: **Comparison of patch reconstruction architectures.** This figure illustrates three different approaches to patch reconstruction. (a) Traditional Encoder/Decoder architecture, where a sparse set of input patches is fed to the encoder, and a small decoder reconstructs all dropped tokens. (b) Our proposed ViDROP architecture, which uniquely combines sparse (dropped) and masked (MSK) tokens in a single encoder. (c) Masked Encoder approach (such as DINOv2 Oquab et al. (2023)) that uses only MSK tokens. In all methods, the input is flattened and patched (image or video), and targets are either pixels or the output of the teacher network (an exponential moving average of the encoder, omitted for clarity).

## 3 METHOD

ViDROP introduces a novel approach to self-supervised learning for video understanding, combining efficient processing with effective representation learning. Our method builds upon the DINOv2 Oquab et al. (2023) framework, adapting it for video data through a sparse encoder architecture that integrates token dropping and masking within a single network. We employ a video sampling and processing strategy using both large and small crops, along with a multi-component loss function ensuring global and local feature consistency. Additionally, we introduce a lossy data loading scheme based on k-means clustering for efficient video compression. In the following subsections, we detail these components and explain how they work together to overcome the limitations of existing video SSL methods.

### 3.1 VIDEO SAMPLING AND PROCESSING

Given a video, we sample multiple clips, creating two large crops and eight small crops. The large crops are fed directly to an exponential moving average (EMA) of the network, serving as the teacher. For the student network, we apply sparsification (token dropping) to these same views and replace some of the remaining tokens with a special MSK token. Small crops maintain dense inputs without sparsification or masking Assran et al. (2022). Both student and teacher networks include a CLS token for global representation.

### 3.2 LOSS FUNCTION

ViDROP's loss function consists of two main components. The first is a DINO-style loss calculated between the CLS tokens of the student and teacher Caron et al. (2021). The second is a patch-level loss for the masked tokens in the student, using the corresponding outputs from the teacher as targets Zhou et al. (2021). We employ a global-global consistency loss to ensure coherence between different views of the same video and a global-local consistency loss that aligns features from large crops (global) with those from small crops (local). The patch-level loss enables fine-grained representation learning, complemented by the KoLeo regularization adapted from DINOv2. For detailed formulations of these losses, we refer readers to the DINOv2 paper Oquab et al. (2023).

Figure 3: **Comparison of video frame compression methods.** Top row: Original frames sampled from the Kinetics-400 dataset Kay et al. (2017). Middle row: Frames compressed and reconstructed using VQ-GAN Esser et al. (2020). Bottom row: Frames compressed and reconstructed using our proposed k-means clustering method Lloyd (1982). Our approach achieves a balance between compression efficiency and visual quality, maintaining compatibility with pretrained models while significantly accelerating data loading.

Unlike traditional consistency-based SSL approaches, we minimize the use of data augmentation Moutakanni et al. (2024), relying primarily on heavy masking and our lossy data compression technique as pseudo-augmentations.

### 3.3 ARCHITECTURE

ViDROP employs a sparse encoder with masked tokens, eliminating the need for a separate decoder (Figure 2). This design differs from asymmetric encoder-decoder architectures He et al. (2021); Tong et al. (2022); Liu et al. (2022); Bardes et al. (2024); Assran et al. (2023); Baevski et al. (2022a) by uniquely combining token dropping and masking strategies within a single encoder. Our approach achieves the efficiency of sparse processing while maintaining the rich representational learning of mask-based self-distillation methods Zhou et al. (2021); Oquab et al. (2023); Bao et al. (2021); Xie et al. (2021); Baevski et al. (2022b). Crucially, our design enables per-patch loss computation, allowing fine-grained representation learning without the computational overhead of dense decoders. This bridges the gap between token dropping efficiency and mask-based representational power.

### 3.4 LOSSY DATA LOADING

To address the data loading bottleneck in video processing, we introduce a lossy data loading scheme based on k-means clustering Lloyd (1982) of video patches (see Figure 3). By applying k-means clustering directly in pixel space on $10 \times 10$ pixel patches, we achieve a compression rate of 150 (using 65536 clusters), significantly accelerating the data loading process.

This approach offers several advantages over VQVAE-based methods van den Oord et al. (2017); Esser et al. (2020); Park et al. (2023); Wiles et al. (2022). It allows for faster processing during both training and inference by avoiding complex encoding and decoding steps. Our method maintains compatibility with pretrained models that operate on raw pixel data, enabling us to leverage existing large-scale pretrained checkpoints. Furthermore, our approach provides a flexible compression scheme that can be easily adjusted to balance between compression rate and representation quality.

## 4 EXPERIMENTS

### 4.1 EXPERIMENTAL SETUP AND PROTOCOLS

We use the training set of Kinetics-400 Kay et al. (2017) for self-supervised training. For evaluation, we use four common action classification datasets (Kinetics-400, Something-Something-

Table 1: **ViDROP ablation experiments.** We report linear probing (single crop) accuracy (%) with ViT-B/16 on K400. If not specified, the default is: the loss is iBOT Zhou et al. (2021), the data augmentation is random resized cropping, the number of small crops is 8, the masking ratio is 85%, and the pre-training length is 60 epochs. Default settings are marked in  gray .

(a) **Loss function.** Patch loss with MASK tokens improves the performance.

| patch loss | linear |
|---|---|
| visible tokens | 51.7 |
| none | 53.1 |
| masked tokens | **54.9** |

(b) **Number of small crops.** Even having a few small crops boosts the performance.

| num. | linear |
|---|---|
| 8 | **54.9** |
| 4 | 54.1 |
| 0 | 50.2 |

(c) **Drop pattern.** Simple random token dropping is more effective than complex patterns.

| pattern | linear |
|---|---|
| random tokens | **54.9** |
| random tubes | 54.5 |
| block (vjepa) | 53.1 |

(d) **Drop rate**. Lower drop rates yield better performance but require longer training times. The model with an 80% drop rate took the longest to train due to reduced batch size.

| rate | time(hh:mm) | linear |
|---|---|---|
| 80% | 31:54 | 55.2 |
| 85% | 24:21 | **54.9** |
| 90% | 23:46 | 54.4 |
| 95% | 23:19 | 51.5 |

(e) **Number of clusters**. Reducing clusters from 65k to 16k maintains accuracy, while 2k clusters slightly decrease it. Using a shared head for 65k clusters also lowers accuracy.

| num. | shared | linear |
|---|---|---|
| 65k | ✗ | **54.9** |
| 16k | ✗ | **54.9** |
| 2k | ✗ | 54.3 |
| 65k | ✓ | 53.8 |

(f) **Masking probability**. Increasing the masking probability range from 10-40% to higher values slightly improves model accuracy (all the models use 16k clusters).

| min | max | linear |
|---|---|---|
| 0.1 | 0.4 | 54.9 |
| 0.1 | 0.7 | 54.9 |
| 0.5 | 0.7 | **55.5** |
| 0.7 | 0.7 | 55.0 |

v2 Goyal et al. (2017), UCF101 Soomro et al. (2012), and HMDB51 Kuehne et al. (2011)) and THUMOS14 Jiang et al. (2014) for temporal action detection. For all evaluations, contrary to common settings in reconstruction-based models Tong et al. (2022); He et al. (2021); Liu et al. (2022); Baevski et al. (2022b); Bardes et al. (2024) but compatible with consistency-based models that rely on heavy data augmentations, we use a **frozen** backbone and only train a linear head on top (except in the case of THUMOS14, where we train a transformer following ActionFormer Zhang et al. (2022)).

For main results, we train ViT-Base, ViT-Large, and ViT-Huge models from pretrained checkpoints. ViT-Base models are trained for 60 epochs with k-means compressed data (24 hours), while ViT-Large and ViT-Huge are trained for 40 epochs without compression (57 and 100 hours, respectively). All use a total batch size of 512 (with gradient accumulation).

For the LEMON Wang et al. (2023b) experiment with random initialization, we follow a progressive schedule: ViT-Small (100 epochs, k-means data, 17 hours), ViT-Base (6 epochs, real data, 12 hours), ViT-Large (19 epochs, 52 hours), and ViT-Huge (30 epochs, 62 hours), totaling 155 epochs and 143 hours (almost 6 days or 1144 V100 hours). For comparison, the VideoMAE ViT-Huge model trained for 1600 epochs is estimated to take approximately 28862 V100 hours ($25\times$ longer).

During training, 85% of the tokens are dropped for the student when fed with two large clips (16 frames of $224 \times 224$). For half of the mini-batch, we randomly mask a portion of the remaining tokens. Additionally, 8 local crops of size $96 \times 96$ (8 frames) are fed as dense input without token dropping or masking.

## 4.2 ABLATIONS

Here we perform in-depth ablation studies on ViDROP design choices with a ViT-B. We use 16384 clusters for the DINO loss, both for the CLS token and the per-patch loss, and mask uniformly between $10\%$ to $40\%$ of the tokens. To accelerate training, we start from a pretrained VideoMAE Tong et al. (2022) checkpoint trained on K400 for 800 epochs.

**Loss function.** Table 1a shows that patch token loss is crucial for ViDROP. An extra loss on masked tokens (iBOT Zhou et al. (2021)) outperforms DINO Caron et al. (2021)'s approach. Using MASK tokens for loss calculation proves more effective than calculating loss on all visible tokens.

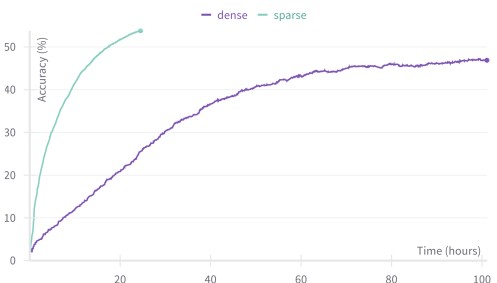

Figure 4: **Probing accuracy as a function of training time for dense and sparse models.** The dense model (without token dropping) takes significantly longer to train and achieves lower accuracy compared to the sparse model (with 85% token drop rate), even after training for four times longer.

Table 2: **Effect of initialization on ViT-Base model performance.** The random model was trained for 240 epochs ($4\times$ the epochs of the pretrained models). Initial pretraining was conducted on 64 Tesla V100 GPUs, and our training on 4 RTX 4090 GPUs.† is estimated by multiplying the training time of the $\text{VMAE}_{1600}$ by the relative training throughput of a ViT-Base and ViT-Large (4.24)

| Setting | Pretraining Time (V100 Days) | Linear Accuracy |
|---|---|---|
| $\text{VMAE}_{800}$ | 74 | 41.0% |
| +ViDROP | +8 | 55.5% |
| $\text{VMAE}_{1600}$ | 148 | 43.5% |
| +ViDROP | +8 | 57.0% |
| $\text{ViDROP}_{\text{rand}}$ | 32 | 53.3% |
| $\text{VMAE}_{1600}^{\text{Large}}$ | $627^{\dagger}$ | 52.5% |

**Number of small crops.** A key success element of DINO Caron et al. (2021) and MSN Assran et al. (2022) was using small crops. We see a similar pattern in Table 1b, where small crops significantly boost performance. We use the same setting as DINOv2 Oquab et al. (2023), but observe that fewer crops are viable. If computational load is an issue (as small crops are not sparsified), the number can be reduced while maintaining considerable performance gains.

**Drop pattern.** While previous video reconstruction methods emphasized specific token dropping patterns Tong et al. (2022); Bardes et al. (2024); Feichtenhofer et al. (2022), Table 1c shows that random patch dropping outperforms tube dropping Tong et al. (2022); Wang et al. (2023a) and block masking Bardes et al. (2024). We observe higher self-supervised loss for these patterns compared to random dropping, suggesting that their difficulty may hinder the model's representational power.

**Drop rate.** Token dropping is an essential component for reducing the computational load of our model. In Table 1d, we can see that there is a trade-off in terms of training time and quality. Reducing the drop rate improves the quality but at the cost of extra training time. Note that in this experiment, we had to reduce the batch size of the model trained with an $80\%$ token drop rate to be able to train it on our hardware. We further studied this trade-off and trained a dense model (i.e., a model with a drop rate of $0\%$) for 100 hours (almost $4\times$ the base model) and achieved an accuracy of only $47.1\%$, which is significantly lower than the $54.9\%$ accuracy of the base model (see Figure 4).

**Number of clusters.** Since we are using Sinkhorn-Knopp centering for our DINO Caron et al. (2021) losses, we are significantly more compute-heavy in the loss (compared to V-JEPA Bardes et al. (2024) and VideoMAE Tong et al. (2022)). Additionally, we can't apply gradient accumulation with many steps, since the centering operation depends on the whole batch. Changing the loss is beyond the scope of this paper, so instead, we reduced the number of clusters both for the global loss and patch loss. Results in Table 1e show that, similar to the findings of MSN Assran et al. (2022), we can significantly reduce the number of clusters and maintain the same performance. We also notice that, similar to the findings of DINOv2 Oquab et al. (2023), having different heads for the two loss terms is beneficial. For the rest of the ablation studies, we used 16k clusters.

**Masking probability.** Following the setting of iBOT Zhou et al. (2021) and DINOv2 Oquab et al. (2023), we apply masking to only half of the large crops. For the other half, we initially randomly masked between 10% to 40% of the remaining tokens (after token dropping). Table 1f shows that we can use larger probabilities and achieve slight improvement. This likely stems from video data's higher redundancy compared to images, allowing for greater sparsity (e.g., 90% for videos Tong et al. (2022) vs 75% for images He et al. (2021)).

**Initialization.** To demonstrate the general applicability of our model, we trained it from scratch and compared it to models initialized with different pretrained weights, as shown in Table 2. The

Table 3: **Comparison of different compression methods.** Evaluation of various methods for compressing and decompressing 0.5M frames on 4 GPUs. $K$Means-based methods offer a good balance between quality (PSNR), processing time, and compression factor.

| Method | PSNR | Time (mm:ss) | Compression Factor |
|---|---|---|---|
| SD-XL | 30.29 | 40:29 | 24 |
| TinyAE-XL$_{byte}$ | 26.44 | 7:25 | 48 |
| VQGAN-f16 | 23.36 | 25:59 | 384 |
| $K$Means$_{16 \times 16}$ | 24.28 | 1:23 | 384 |
| $K$Means$_{10 \times 10}$ | 25.75 | 1:41 | 150 |

Table 4: **Effect of $K$Means on training speed and accuracy with ViT-Small.** Using $K$Means compression achieves a $5.37\times$ speedup with a minor performance cost. Training on pixel data for the same duration results in lower performance.

| $K$Means | Time (hh:mm) | Linear Accuracy |
|---|---|---|
| ✓ | 9:42 | 44.4 |
| ✗ | 52:09 | 46.1 |
| ✗ | 9:42 | 29.5 |

Table 5: **Training throughput of various SSL methods with ViT models.** Comparison of training speeds for different SSL methods using ViT-Base, ViT-Large, and ViT-Huge architectures on a single NVIDIA RTX 4090 GPU. DINOv2 refers to the dense version of our model without sparsity. While ViDROP shows lower raw throughput compared to reconstruction-based methods, it significantly outperforms consistency-based models (DINOv2 and SVT) and achieves superior sample efficiency (see Table 2).

| Method | ViT-Base | | ViT-Large | | ViT-Huge | |
|---|---|---|---|---|---|---|
| | Max B.S. | Throughput | Max B.S. | Throughput | Max B.S. | Throughput |
| SVT | 4 | 8.4 | 1 | 1.5 | OOM | N/A |
| VMAE | 30 | 139.6 | 8 | 32.9 | 4 | 17.0 |
| VMAEv2 | 15 | 58.8 | 8 | 29.7 | 4 | 16.1 |
| VJEPA | 37 | 74.4 | 22 | 37.9 | 8 | 16.2 |
| DINOv2 | 12 | 23.4 | 4 | 7.8 | 1 | 3.1 |
| ViDROP | 33 | 43.1 | 11 | 17.4 | 4 | 7.8 |

randomly initialized model was trained for $4\times$ the epochs of the pretrained models but still required significantly less total training time when considering the pretraining duration of the checkpoints.

**Data compression method.** To accelerate training for ViT-Small and ViT-Base models, we employed $K$Means-based data compression. Table 3 shows that $K$Means compression strikes a good balance between quality (measured by PSNR), encoding/decoding time, and compression rate. Other methods rely on large models, complicating inference and disallowing the use of pretrained checkpoints. For all ablation experiments, data was compressed once, taking around 40 hours for 60 epochs of data. We used a patch size of $10 \times 10$ and 65536 clusters.

Table 4 demonstrates that using $K$Means data results in a $5.37\times$ speedup with a small performance cost when training a ViT-Small model. Training the same model on pixel data for the same duration yields worse performance. ViT-Base sees a $2.71\times$ speedup, and ViT-Large a $1.24\times$ speedup (better GPUs can lead to greater speedup, as data becomes the bottleneck).

**Training throughput.** Table 5 compares the training speed of various self-supervised learning (SSL) methods for video ViT models on a single NVIDIA RTX 4090 GPU (24 GB VRAM). We used official code and configurations for each model: VJEPA Bardes et al. (2024) with repeated masking of 2, VideoMAEv2 Wang et al. (2023a) with 4, and VideoMAE Tong et al. (2022) without repeated masking. DINOv2 represents the dense version of our model without sparsity. Measurements exclude data loading time, focusing on forward and backward passes. While ViDROP shows lower raw throughput compared to reconstruction-based methods due to its combination of small and large crops and more complex loss calculation, it significantly outperforms consistency-based models like SVT and DINOv2. Crucially, as demonstrated in Table 2, ViDROP achieves superior sample efficiency, reaching higher performance with fewer training iterations, thus offsetting its lower per-iteration speed.

Table 6: **Comprehensive performance comparison on action classification tasks.** We report linear probing accuracies (%) for all datasets and KNN accuracies for smaller datasets. Numbers in parentheses indicate evaluation clips. For K400, we also present low-shot learning results with varying amounts of labeled data. For temporal action detection on THUMOS14, we report the average mean Average Precision (mAP) across different temporal Intersection over Union (tIoU) thresholds. ViDROP variants consistently outperform their baselines across all tasks and metrics. Superscript numbers indicate performance gap compared to the respective baseline.

| Method | K400 | | SSv2 | UCF-101 | | HMDB-51 | | K400 Low-shot | | | THU. |
|---|---|---|---|---|---|---|---|---|---|---|---|
| | Linear (5×3) | Attn. (1×1) | Linear (2×3) | Linear (5×3) | KNN (1×1) | Linear (5×3) | KNN (1×1) | 5% | 10% | 50% | Avg. mAP |
| | | | | | | | | Linear (5×3) | | | |
| $\rho$BYOL | 71.5 | N/A | 25.3 | 89.6 | 85.2 | 61.2 | 49.7 | - | - | - | - |
| SVT | 68.1 | N/A | 20.3 | 91.3 | 87.2 | 63.1 | 51.8 | - | - | - | - |
| $\text{VMAE}_{1600}^{\text{Large}}$ | 60.7 | 68.6 | 27.9 | 84.5 | 49.1 | 60.3 | 29.2 | 43.3 | 50.4 | 59.6 | 15.0 |
| +ViDROP $_{\text{noaug}}^{\text{kmeans}}$ | $63.4^{+2.7}$ | $71.9^{+3.3}$ | $32.9^{+5.0}$ | $86.6^{+2.1}$ | $73.0^{+23.9}$ | $60.9^{+0.6}$ | $45.5^{+16.3}$ | $50.4^{+7.1}$ | $55.8^{+5.4}$ | $64.7^{+5.1}$ | $50.2^{+35.2}$ |
| +ViDROP | $74.3^{+13.6}$ | $72.6^{+4.0}$ | $38.4^{+10.5}$ | $\mathbf{94.2^{+9.7}}$ | $\mathbf{88.0^{+38.9}}$ | $\mathbf{69.6^{+9.3}}$ | $55.2^{+26.0}$ | $59.7^{+16.4}$ | $64.2^{+13.8}$ | $71.8^{+12.2}$ | $\mathbf{57.1^{+42.1}}$ |
| $\text{VJEPA}^{\text{Large}}$ | 62.7 | **73.7** | 43.2 | 92.0 | 81.2 | 66.9 | 54.7 | 49.6 | 54.8 | 58.0 | 20.1 |
| +ViDROP $_{\text{noaug}}$ | $72.4^{+9.7}$ | $71.1^{-2.6}$ | $33.8^{-9.4}$ | $91.1^{-0.9}$ | $81.3^{+0.1}$ | $66.0^{-0.9}$ | $51.0^{-3.7}$ | $58.5^{+8.9}$ | $62.5^{+7.7}$ | $69.8^{+11.8}$ | $49.9^{+29.8}$ |
| +ViDROP | $\mathbf{74.8^{+12.1}}$ | $72.7^{-1.0}$ | $38.7^{-4.5}$ | $93.7^{+1.7}$ | $84.8^{+3.6}$ | $\mathbf{69.6^{+2.7}}$ | $\mathbf{55.3^{+0.6}}$ | $\mathbf{61.2^{+11.6}}$ | $\mathbf{65.4^{+10.6}}$ | $\mathbf{72.5^{+14.5}}$ | $56.2^{+36.1}$ |
| $\text{VMAE}_{1600}^{\text{Huge}}$ | 67.2 | - | 29.3 | 84.3 | 48.6 | 59.7 | 30.1 | 46.8 | 52.9 | 64.5 | 41.7 |
| +ViDROP | $\mathbf{74.8^{+7.6}}$ | - | $\mathbf{39.3^{+10.0}}$ | $\mathbf{94.2^{+9.9}}$ | $85.9^{+37.3}$ | $69.4^{+9.7}$ | $53.4^{+23.3}$ | $60.3^{+13.5}$ | $64.4^{+11.5}$ | $72.2^{+7.7}$ | $55.7^{+14.0}$ |
| 🍋ViDROP $_{\text{noaug}}^{\text{Huge}}$ | $66.3^{-0.9}$ | - | $30.1^{+0.8}$ | $85.2^{+0.9}$ | $75.7^{+27.1}$ | $59.6^{-0.1}$ | $45.4^{+15.3}$ | $50.1^{+3.3}$ | $55.4^{+2.5}$ | $63.7^{-0.8}$ | $47.2^{+5.5}$ |

## 4.3 RESULTS

For our main results, we train seven different models to comprehensively evaluate the performance of ViDROP under various conditions. We begin with two ViT-Large models based on VideoMAE Tong et al. (2022), one with minimal data augmentation and another incorporating heavier augmentations used in DINO Caron et al. (2021). These are followed by two similarly configured ViT-Large models based on VJEPA Bardes et al. (2024), demonstrating the compatibility and versatility of our method across different pretraining paradigms. To showcase scalability, we include two ViT-Huge models: one initialized from a VideoMAE checkpoint and another trained using LEMON Wang et al. (2023b) expansion techniques from random initialization, highlighting the efficiency and potential of our method for training large-scale models with limited computational resources. All VideoMAE-based models are initialized with checkpoints pretrained for 1600 epochs on Kinetics400, while VJEPA-based models use the original VJEPA checkpoint. For one VideoMAE-based ViT-Large model, we utilize $K$Means compressed data. While $K$Means compression doesn't offer significant speed benefits for ViT-Large models, we include this configuration to demonstrate the robustness of our findings. By using compressed data, we establish a lower bound on performance (as shown in Table 4), yet still outperform the original VideoMAE model, underscoring the effectiveness of ViDROP even under potentially suboptimal data conditions. Additionally, to facilitate direct comparisons with academic papers, we separately train a ViT-Base model based on VideoMAE trained for 800 epochs on Kinetics400.

**Action classification results.** Table 6 presents our comprehensive results on various action classification tasks, including standard action recognition, low-shot learning, and temporal action detection. For action recognition, ViDROP variants consistently improve linear probing accuracies across all datasets, with the exception of VJEPA-based models on some datasets due to potential forgetting effects from extra training data of VJEPA Bardes et al. (2024). Notably, attention probing for VideoMAE-based models shows improvement, while VJEPA-based models exhibit lower attention probing accuracies compared to linear probing, reminiscent of DINOv2's behavior on ImageNet. KNN accuracies see substantial improvements, particularly for VideoMAE-based models. The incorporation of data augmentation further enhances performance across metrics. In low-shot settings on K400, our ViDROP models often match or surpass the full data baseline using only 10% of the labeled data, demonstrating strong data efficiency. For temporal action detection on THUMOS14 Jiang et al. (2014), ViDROP shows remarkable improvement over baselines, with the average mAP more than doubling, indicating more temporally precise feature representations. The LEMON-trained model, starting from random initialization, achieves comparable results to

Table 7: **Performance comparison of ViT-Base models.** Linear probing accuracies (%) on various datasets, DAVIS tracking $((\mathcal{J}\&\mathcal{F})_m$ metric), and copy detection ($\mu$AP). SIGMA$_{DINO}$ Salehi et al. (2024) uses a teacher model pretrained on extra data.

| Method | SSv2 | K400 | UCF | HMDB | IN-1K | DAVIS | Copy Det. |
|---|---|---|---|---|---|---|---|
| SIGMA$_{MLP}$ | 19.9 | 30.7 | 73.8 | 45.0 | 24.1 | - | - |
| SIGMA$_{DINO}$ | 20.8 | 47.5 | 80.7 | 52.3 | **45.0** | - | - |
| VMAE$_{800}^{Base}$ | 17.5 | 20.7 | 58.6 | 37.7 | 20.2 | 40.8 | 2.0 |
| +ViDROP $_{noaug}$ | **25.4** | **59.5** | **81.7** | **54.6** | 44.6 | **54.8** | **22.4** |

VMAE$_{1600}^{Huge}$, with slightly lower performance on full K400 but improved results in low-shot scenarios, highlighting the potential of progressive training for large-scale video SSL.

These comprehensive results demonstrate the effectiveness and scalability of ViDROP across various model sizes, datasets, and action understanding tasks. Our method consistently achieves state-of-the-art performance in linear probing, shows strong transferability to smaller datasets and low-shot scenarios, and exhibits remarkable generalization to temporal action detection, underscoring its potential for a wide range of video understanding applications.

**Comparison with small scale methods.** To facilitate fair comparisons with small scale approaches, we trained a ViT-Base model following the setting of SIGMA Salehi et al. (2024) (current SotA method in small scale models). Table 7 presents the results of this comparison across various datasets and tasks. For ImageNet (IN-1K) evaluation, we follow the common practice of repeating each image twice to create a pseudo-video input. Our method significantly outperforms both SIGMA variants and the VideoMAE baseline across all datasets, demonstrating its effectiveness even without additional data augmentation. Notably, ViDROP achieves comparable or superior performance to SIGMA$_{DINO}$, which benefits from a teacher model pretrained on extra imagenet data. The results also showcase ViDROP's versatility on DAVIS tracking Pont-Tuset et al. (2017) and copy detection Pizzi et al. (2023) tasks. For DAVIS tracking, we adopt the DINO approach of processing frames individually. The findings highlight ViDROP's substantial improvements over VideoMAE baselines, particularly in copy detection where ViDROP achieves a remarkable 22.4 $\mu$AP, compared to 2.0 for the VideoMAE baseline. These results underscore ViDROP's ability to learn rich, transferable representations across diverse video understanding tasks.

## 5  CONCLUSION

We presented ViDROP, a novel self-supervised learning approach for video understanding that combines token dropping and masking strategies within a single encoder. Our method achieves state-of-the-art performance across various video understanding tasks, including action recognition, temporal action detection, and low-shot learning, while maintaining computational efficiency. Key innovations include a sparse encoder architecture that eliminates the need for a separate decoder, enabling fine-grained representation learning without computational overhead, and a lossy data loading scheme based on k-means clustering, significantly accelerating data processing while maintaining compatibility with pretrained models. We demonstrated scalability across model sizes and initialization strategies, including successful training of ViT-Huge models from scratch using limited computational resources. ViDROP's remarkable data efficiency in low-shot learning scenarios and its effectiveness in both action classification and temporal action detection tasks highlight its versatility and potential for real-world applications.

Looking forward, our work opens up possibilities for training state-of-the-art models with limited computational resources, making advanced video understanding more accessible. However, our results with VJEPA-initialized models highlight the importance of training on more diverse datasets to enhance generalization capabilities. Additionally, while our LEMON experiments for training ViT-Huge models from scratch show promising results, they are not yet on par with models initialized from pretrained weights, indicating a need for further exploration of optimal scaling strategies.

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
