# OpenReview forum: "ViDROP: Video Dense Representation through Spatio-Temporal Sparsity"
_ICLR.cc/2025/Conference — ICLR 2025 Conference Withdrawn Submission_

### Official Review · Reviewer_37vw · 2024-11-02

**Soundness:** 1
**Presentation:** 2
**Contribution:** 2
**Rating:** 5
**Confidence:** 3

**Summary:**

The paper discusses a novel architecture named ViDROP (Video Dense Representation thOugh  spatio-temporal sparsity). Essentially, it uses token masking and token dropping strategy together to overcome limitations of the previous SSL architectures. The paper also makes use of the k-means clustering technique to load the data which speeds up the training. The paper reports results across various ViT model sizes and on various video datasets (e.g. K400, SSv2, UCF101, HMDB51 and THUMOS14).

**Strengths:**

The abstract and introduction nicely written. The idea of the paper is simple enough and apparently seems to work.

**Weaknesses:**

The main drawback for me is the way the rest of the paper (apart from abstract and introduction) is written. It is mostly confusing, and sometimes misleading and missing a lot of details that should have been included to answer the questions of the readers. I will give plentiful examples for this in the rest of the review.

**Questions:**

Starting with the introduction, two motivations for the problem statement are mentioned in the lines 48 to 53. It is the second motivation "can lead to suboptimal representation learning...." till end of the line 053 that is not well supported by any citations at all. If this is true, it must have been proven through previous research on this specific topic which must be cited here.

Although the main topic of the paper is SSL on Videos, the paper does not do justice to this in prior work. I would expect to get a comprehensive picture of Video SSL methods and how ViDROP is different from each of those and how does ViDROP fit into overall SOTA...

Similarly, there are problems present in the writing of methods section which add confusion to what exactly the technique is doing. For example, in section 3.1 (line 200), the paper mentions "given a video, we sample multiple clips" Multiple clips? How many exactly? Is this followed some benchmark or protocol introduced by some other work?
In the very next lines (201 - 203), it is not mentioned how/from where the teacher is initialized.
Similarly in the section 3.2, there is no mention of what mathematical function does the loss take and how is it working to achieve the desired objective?
Section 3.4 can also be polished in terms of implementation of the technique on the dataset.

Experimental section 4, also lacks clarity.  In Line 269, it is not clear if the models are initialized from checkpoints or being trained from scratch. Lines 301-304 are quite opaque. It's not clear which pretrained checkpoints, what does 24 hours of compressed data even mean here. The LEMON technique is not introduced in a sentence or two and it is left to the reader to find it out.
Line 304 does not tell how many gradient accumulation steps?
Which protocol is being followed in Line 310-313?

Although the ablations seem to be nice, but upon looking closely, they seem to be lacking important details. In line 320, VideoMAE checkpoint is mentioned for training. It is not mentioned what is the checkpoint further trained on...
In line 363, compute-heavy is not justified by any reference or ablation. Number of Gradient accumulation steps are not mentioned.
Line 369 is also problematic as it implies if 16k clusters settings is changed or not?

Line 377, initialization section is also confusing. Not clear what dataset the network is being trained on.
Line 416, not clear if the compression changes distribution or not. And if the validation dataset is being compressed or not. And what does it mean 60 epochs of data?
Line 427, how much time does it take to load data?
The main focus of the paper is on the introduction of sparsity and dropping. This has been touched upon table 1(d) and f but it does not cover enough range to show the performance with and without these factors at all. For example, drop rate from 0 to 100% in steps of 10% would be nice to show the complete picture. Likewise for masking probability.

Minor:
The focus of the paper is not entirely clear whether the speed gains are coming from the introduction of clustering technique or the token dropping and masking technique. Citations are missing.
Table 1 (e), 16k cluster should be grayed out since that's the setting being used, right?
The paper lacks visuals and graphs for explanation of the different concepts and graphs for ablations,.

Conclusion:
Overall the paper is nice and I would have accepted the paper if there were not significants problems in the writing adding the to confusion, opaqueness of the whole approach and adding doubts to the reproduction of the resutls. All of these changes add up and warrant rewriting of the paper which will be a major revision and therefore I would lean towards rejecting the paper unfortunately.

---

### Official Review · Reviewer_Vw9N · 2024-11-03

**Soundness:** 2
**Presentation:** 2
**Contribution:** 1
**Rating:** 3
**Confidence:** 4

**Summary:**

This paper introduces a novel self-supervised learning approach, i.e. ViDROP, by combining masked token modeling and token compression techniques. ViDROP demonstrates superiority in various video understanding tasks.

**Strengths:**

1.	ViDROP removes the decoder network and only adopts an encoder for learning visual feature representation, saving computational costs.
2.	Fine-grained, patch-level correspondence is learned through the local-global consistency loss, going beyond traditional global feature consistency.

**Weaknesses:**

1.	In Table 1(d) and lines 358-359, the experiments should be conducted on the same batch size while altering the drop rate, to mitigate the impact of the batch size on training efficiency (and perhaps the final performance). Additionally, the authors found that the model trained on full tokens (0% drop rate) achieves significantly lower performance than the base model. In this case, the authors are encouraged to study more drop ratios, e.g. which ratio leads to the best performance.
2.	It is not very clear how K-means works in Section 3.4, what is the actual input to the model after clustering?
3.	The authors claimed that K-means accelerates the data loading process. The comparison of data loading time as well as the training time should be reported. As a fair comparison, the efficiency and performance of training on randomly selected image patches (vs K-means) are not reported.
4.	Regarding action recognition, the authors are encouraged to additionally report the full fine-tuning results.

**Questions:**

1.	What is the pre-trained checkpoint in Line 301?
2.	Why ViT-B is trained with k-means compressed data while ViT-L/H are trained without compression?
3.	In line 310, what is the masking ratio of the remaining masked tokens? How does this masking ratio affect the efficiency and final performance?
4.	In lines 147-148, the authors claim that AVION remained relatively slow. Is there any evidence? According to their paper, AVION offers significant improvement in training efficiency by speeding up the time-consuming video resizing operation during data loading.

---

### Official Review · Reviewer_cyiN · 2024-11-08

**Soundness:** 3
**Presentation:** 2
**Contribution:** 3
**Rating:** 5
**Confidence:** 4

**Summary:**

This paper focuses on self-supervised learning (SSL) for video representation. The main contribution of the paper is an efficient SSL method that combines the token dropping strategy in MAE and masked token reconstruction in DINOv2, with a combination of loss functions (e.g., DINO-style loss, patch-level loss) and training strategies (e.g., using small-crop samples) that have been shown effective for image/video SSL. Another contribution is to propose a K-Means clustering based video compression method that help improve the efficiency in data loading. The method is efficient due to the design of token dropping and compressed video input, and the experiments verify its effectiveness -- the method can provide further improvement over pretrained models on common video action recognition benchmarks.

**Strengths:**

The main idea of the paper is simple and intuitive -- marrying the training strategies from MAE and DINOv2 potentially provides an efficient and effective method for video SSL. The paper shows good execution on implementing the ideas and provides extensive ablation study to verify the effectiveness of each design choice. The final experiment results (as shown in Table 5) are positive overall, showing that more effective video representation is learned by the proposed SSL method.

**Weaknesses:**

1. The writing of the paper needs substantial improvement, many technical details are missing or not clear, and some experiment results are not explained.

1.1. What are the global-local consistency loss and global-global consistency loss in L212 -- L213? If the losses are used in DINOv2, maybe it's more clear to specify the equation number or the corresponding name in DINOv2.

1.2. In L303, why the ViT-L and ViT-H are trained without compression at all? Is it because video compression provides less speedup when larger models is used (data loading is not a bottleneck anymore). I'd suggest the authors to describe the experimental setup and details more clearly.

1.3. From L311 and L372, I don't really understand the difference of the masking strategies between the two halves, and the reason (or ablation) for using different strategies is not explained.

1.4. Table 1f shows that better results can be obtained by using (min=0.5, max=0.7), and the improvement is even more significant than those shown in other ablation. Why not using this as the default setting?

1.5. It's not clear how the K-Means based video compression method provides speedup for data loading. Is it due to the lower resolution when compressing each 10x10 patch? If so, it'll be also interesting to compare the compression quality (PSNR) with some simple strategies such as taking the mean or median for each 10x10 patch.

1.6. In Table 6, significant performance drop is observed for K400-Attn. and SSv2-Linear settings. The paper doesn't provide any explanation or analysis to these two important results.


2. It is not necessarily a key concern but overall the novelty of this paper is limited because most of the training strategies are proposed in prior work such as MAE and DINOv2.

**Questions:**

Please find some of the questions in the Weakness section.

More questions:
1. Could you provide some insights or analysis why ViDROP provide more significant gains on the temporal action detection task on THU?
2. The Atten-pool results on SSv2 is missing. Could you provide the results for completeness and compare with prior work like VJEPA?

---

### Official Review · Reviewer_JoXn · 2024-11-09

**Soundness:** 3
**Presentation:** 2
**Contribution:** 2
**Rating:** 3
**Confidence:** 4

**Summary:**

This paper presents a self-supervised video representation learning framework, termed as ViDROP. It combines the existing techniques of dropping and masking to speed up the pre-training efficiency. The pre-training pipeline and loss functions are very similar previous methods such as DINO v2. It performs experiments with backbones from ViT-Based to ViT-Large on the datasets of Kinetics, SSth V2, UCF101, HMDB51, and THUMOS, to verify the effectiveness of its proposed framework. Meanwhile it also performs experiments on small datasets. The experiment results indicate the effectiveness of proposed SSL framework.

**Strengths:**

+ Self-supervised video representation is an important problem in computer vision. This paper presents an good exploration to this direction.

+ The evaluation is relatively comprehensive to demonstrate the effectiveness of proposed SSL framework. The performance seems to be very good within the proposed framework.

**Weaknesses:**

- It seems the technical contribution of this paper is limited. The proposed  dropping and masking strategies are both existing techniques. The combination of them is also very straightforward. Using the dropping strategy to speed up the pre-training process of contrastive learning is not new, and has been studied in the following paper.

- The whole pre-pretraining pipeline and loss functions are very similar to DINO v2 and the description in Section 3.2 is very short, indicating the techinical novelty is limited.


- The technical description is very short and some remains unclear: For example, the illustration in Figure 2 is not clear and can not tell how to calculate the training loss. Meanwhile, it is unclear how to model the connection between the large crops and small crops. It is suggested to provide a clear and detailed pipeline figure to illustrate the whole pre-training process.  Second, the motivation and description on lossy data loading is unclear. The very short description in Section 3.4 is really hard to find the implementation details.

- About evaluation, although linear probing evaluation with frozen backbone is very efficient, the full fine-tuning is more effective and can test the up-bound performance of the pre-trained model. Meanwhile, the linear probing evaluation is more friendly to consistency-based method than reconstruction methods. The contrastive loss is a kind of discriminative objective and encourges the learned representation to be more sepearable for classification. But, the reconstruction loss is only for pixel prediction and is less effective for classification if the pre-trained is frozen. So, I think we should test and compare the effectivness of learned representation in both linear probe setting (for generation power) and full fine-tuning setting ( for its performance limit).

- The performance comparison in Table 6 is confused. The performance drop against VJEPA is very large on the SSv2 dataset. Is there any reason behind this result? Meanwhile, from ViT-Large to ViT-Huge, the performance improvement is very small on the Kinetics dataset.

- The proposed method is not independent SSL method and its good performance is still dependent on the VideoMAE pre-trained or VJEPA pre-trained models. It is really important to improve the performance without initialization from previous checkpoints.

**Questions:**

Why k-means data compression is applied to ViT-Based, rather than ViT-Large or ViT-Huge?

Following the previous VideoMAE work, it is suggested to report the action detection performance on the AVA dataset.

---

### Official Review · Reviewer_UZTt · 2024-11-09

**Soundness:** 3
**Presentation:** 2
**Contribution:** 4
**Rating:** 6
**Confidence:** 4

**Summary:**

The paper proposed to use token dropping and mask strategy for video self-supervised learning in an encoder-only architecture. Such an approach brings the effectiveness of token dropping but without the use of an expensive decoder supervised using pixel-reconstruction objective. The paper shows the effectiveness of the approach using linear probes over various downstream video tasks.

**Strengths:**

- The direction of research looks encouraging as it pools the advantages of VideoMAE and DINO in a single framework
- The paper goes beyond action classification to show the effectiveness of the approach on action detection and object tracking

**Weaknesses:**

### Apples-to-apples comparison in main results (Sec 4.3 and Table 6)
- The paper specifies the use of VideoMAE checkpoints at 1600 epochs as initialization (lines 464-466), and use them as baselines to report performance improvements in Table 6
- However, since the proposed approach performs additional training on top of these checkpoints, it is not clear whether these comparisons are fair
- Couple of ways the comparison can be made apple-to-apples is
    - to compare VideoMAE(1600ep) with VideoMAE(800ep) + ViDROP(800ep).
    - or training VideoMAE(1600ep) for the additional number of epochs that the paper uses to train ViDROP
- The authors listed a strategy to scale model size from scratch in lines 305-309, and provide results for ViT-Huge variant in Table 6. Do the authors have the results for ViT-Large trained from scratch?

### Is linear probing the best way to evaluate VideoMAE features?
- The authors consistently use linear probing to draw comparisons with VideoMAE, but only fine-tuning is used in their paper [1]. Did the authors try fine-tuning?


### Naive baseline
- In Figure 4, the authors made a comparison with a variant of the model with no token dropping, which is close to DINOv2 approach (Fig 2c), and shows that it converges slower
- On the other hand, the approach also address the limitation of VideoMAE-style encoder-decoder approach (Fig 2a). One baseline that could be included is to use the same drop+mask strategy but reconstruct mask tokens using a decoder. This would help draw direct comparisons with VideoMAE architecture and give readers some insights.
- Moreover this would highlight whether the improvement is brought by the use of additional losses or model architecture

### Lack of notations and uniform use of terminology
- Table 1 caption (line 272) specifies a “masking ratio is 85%” but lines 319-320 mentions “mask uniformly between 10% to 40% of the tokens”. This creates a confusion between mask and drop tokens, and readers are left to parse the details.
    - Notations for drop and mask ratios could be introduced in the text to as done in previous works such as VideoMAEv2 [1] ($\rho$)
- Further, tokens that are masked are denoted as “MSK” in Figure 2 and lines 203-204, but later denoted as “MASK” in Table 1 and line 323

- In Table 1, although 85% is desirable w.r.t. accuracy and time trade-off, the accuracy in bold should be the first row (for 80%)

### Additional details (such as supplementary material) would have been useful
- A model card containing complete hyperparameter setting for the experiments and details on model architecture. For example, VideoMAEv2 [1] shared details of model architecture along with pre-training, post pre-training, and fine-tuning settings
- Details regarding k-means clustering, and how it is done on 10x10 pixel patches (lines 255-257) would be useful


### Justification
The paper shows strong performance of the approach on various benchmarks, but it is not clear whether we should favour the proposed approach over VideoMAE or DINO line of research. The results seems to lack head-to-head comparisons, use of naive baselines, and additional details to verify the claims and draw conclusions.

Please note
- I have not checked the accuracy of the claim that linear probing results are state-of-the-art
- And I did not fully understand the k-means clustering details mentioned in lines 255-257


### References
[1] Wang, Limin, et al. "Videomae v2: Scaling video masked autoencoders with dual masking." In CVPR 2023.

**Questions:**

please see weaknesses

---

### Note · Authors · 2024-11-15

I have read and agree with the venue's withdrawal policy on behalf of myself and my co-authors.